# A dynamical adjustment perspective on extreme event attribution

Laurent Terray[1]

[1]CECI, Université de Toulouse, CERFACS/CNRS, Toulouse, France

*Correspondence to*: Laurent Terray (terray@cerfacs.fr)

5 **Abstract.**

Here we demonstrate that dynamical adjustment allows a straightforward approach to extreme event attribution within a conditional framework. We illustrate the potential of the approach with two iconic extreme events that occurred in 2010: the early winter European cold spell and the Russian summer heat wave. We use a dynamical adjustment approach based on 10 constructed atmospheric circulation analogues to isolate the various contributions to these two extreme events using only observational and reanalysis datasets. Dynamical adjustment results confirm previous findings regarding the role of atmospheric circulation in the two extreme events and provide a quantitative estimate of the various dynamic and thermodynamic contributions to the event amplitude. Furthermore, the approach is also used to identify the drivers of the recent 1979–2018 trends in summer extreme maximum and minimum temperature changes over western Europe and western 15 Asia. The results suggest a significant role of the dynamic component in explaining temperature extreme changes in different regions, including regions around the Black and Caspian Seas as well as central Europe and the coasts of western Europe. Finally, dynamical adjustment offers a simple and complementary storyline approach to extreme event attribution with the advantage that no climate model simulations are needed, making it a promising candidate for the fast-track component of any real-time extreme event attribution system.

20 **1 Introduction**

Extreme weather events such as heat waves and cold spells have a profound impact on human health (Guo et al. 2018; Robine et al. 2008), natural ecosystems (Stillman 2019), social systems and economy (Jahn 2015). Europe has experienced a high number of extreme temperature episodes since the early 2000s. Recent examples include the summer 2003 heat wave over western Europe, the summer 2010 heat wave over eastern Europe and Russia, the 2010 cold winter over Europe, the 25 2012 cold spell over eastern and northern Europe, the summer 2015 heat wave over southern and central Europe and the summer 2018 heat waves over North-western and Central Europe. Science questions related to the origin, causal and amplifying factors as well as predictability and prediction of these events have led to an unprecedented number of studies in the last 20 years, with 2003 being perhaps the starting point of this intense wave of research activity (Stott et al., 2004). This

emerging field of research is often referred as extreme event attribution, although it often covers a range of questions and
issues that go beyond the standard attribution framework (Hegerl et al., 2011; Lloyd and Shepherd, 2020).

Recent and exhaustive review papers have nicely summarized the multiple modelling and statistical approaches and framings
that have been used in the field of extreme event attribution (Stott et al., 2016; Shepherd 2016; Otto 2017; Naveau et al.,
2020). A first type of approach, (from now on the risk-based approach) is focusing on estimating and comparing the
frequency of occurrence of extreme events under two stationary worlds, the factual one (with the effect of human influence
on climate) and the counterfactual one (with no human influence on climate). A second type of approach (thereafter the
process-based approach) puts more emphasis on the identification of the physical drivers of extreme events.  Within this
second approach, the main objective is to quantify the influence of the key causal factors of the extreme event under scrutiny
rather than estimating changes in the likelihood of the event due to human influence (see Wehrli et al., 2019 for a perfect
example of the process-based approach). Both risk- and process-based approaches can often be combined in some ways to
improve the understanding and robustness of extreme event attribution results (Otto et al., 2012). Note that the process-based
approach can also be viewed as a sub-category of the storyline approach that focuses on the key drivers and physically
plausible unpacking of past events (Shepherd et al., 2018).

Within the process-based approach, the quantification of the driver's influence often relies upon model sensitivity
experiments to disentangle the impact of each causal factor. Different modelling frameworks can be used (Schär and Kröner,
2017; Wehrli et al., 2019): the first one is based on "all-but-one" experiments where the influence of one specific factor is
removed from the control simulation setup (here control simulation means a simulation including the influence of all
factors). The second one, based on "only-one" experiments, goes in the other direction by accounting for the influence of a
specific causal factor in a control simulation (with all other factor's influence removed).

A subset of the process-based approach uses the fact that the vast majority of extreme events (in particular at mid-to-high
latitudes) are associated with specific (but not necessarily extreme) atmospheric circulation patterns. Conditioning the
observed temperature or precipitation extreme variations on the appropriate circulation pattern naturally leads to decompose
the extreme event characteristics (such as amplitude and persistence) into dynamic and thermodynamic components. As the
two components have very different signal-to-noise ratios related to the response to anthropogenic forcing, extreme event
attribution results can be notably strengthened by focusing separately on the two aspects (Shepherd, 2016; Vautard et al.,
2016).
The above decomposition can easily be performed based on atmospheric circulation nudging experiments for different mean
climate states corresponding to contrasted values of the thermodynamic drivers (either external forcings or internal
variability factors). For instance, Wehrli et al. (2019) quantify the influence of sea surface temperatures (SSTs) and soil
moisture to five recent heatwaves in both subtropical and extratropical regions using global atmospheric simulations with

atmospheric circulation nudged to reanalysis (using grid-point nudging). Based on nudged regional model experiments, Meredith et al. (2015) have shown that Black Sea SST recent warming has been a key contributor and amplifier in the magnitude of the Krymsk July 2012 precipitation extreme. Another recent example about heat wave attribution is the application of a methodology based on spectral nudging of the free atmosphere within a global model and applied to both factual and counterfactual worlds (van Garderen et al., 2021).

An alternative approach to model-based studies is to apply a dynamical adjustment diagnostic approach to observations and/or reanalyses. Dynamical adjustment methods have initially been developed to illustrate and quantify the role of atmospheric internal variability on long-term temperature regional trends (Wallace et al., 2012; Smoliak et al., 2015; Guan et al., 2015; Deser et al., 2016; Saffioti et al., 2016; Gong et al., 2019; Sippel et al., 2019). They have also been applied in other contexts such as attribution studies of regional precipitation changes (Guo et al., 2019; Lehner et al., 2018), time of emergence uncertainties (Lehner et al., 2017), influence of low-frequency oceanic modes on continental climate (O'Reilly et al., 2017) and land-atmosphere interaction studies (Merrifield et al., 2017). The dynamical adjustment method pioneered in Deser et al. (2016) is based on the constructed analogue approach and was initially applied using monthly mean sea level pressure and temperature fields.

Here we investigate the possible added value of the constructed analogue dynamical adjustment approach in identifying and disentangling the key drivers and related physical processes of extreme events. We first use dynamical adjustment to assess the contribution of atmospheric circulation and other drivers to two specific and iconic extreme events: the 2009–2010 cold European winter (Wang et al., 2010; Cattiaux et al., 2011; Osborne 2011) and the 2010 Russian heat wave (Barriopedro et al., 2011; Dole et al., 2011). The analysis is performed at daily time scales for the two events allowing to yield insights on both the chronology and time-mean aspects. One key advantage of the dynamical adjustment approach is that it can be used with observational (and/or reanalyses) data without the need of additional atmospheric (or climate) model simulations. One limitation is that using dynamical adjustment with only observations does not allow to make statements regarding the role of any particular external forcing (for instance greenhouse gases or aerosols). Note that this inability to make single-forcing attribution statements does not come from the dynamical adjustment itself but rather from the fact that the approach used in this work only relies on observations. Indeed, dynamical adjustment could also be used on large ensembles of single-forcing simulations such as those presented in Deser et al. (2020) or the ones performed under the DAMIP framework (Gillett et al., 2016).

Observational uncertainty estimates can be derived by using multiple products and/or perturbed-parameter observational ensemble. Uncertainty related to the dynamical adjustment method parameters can be estimated by adequate sampling of the latter. Finally, the approach can be used for any type of event as long as high-quality observational daily datasets of both

atmospheric circulation and the physical variable of interest are available for a sufficiently long common period (at least 30 years).

The second objective of this study is to revisit the attribution of the links between recent changes in atmospheric circulation
patterns and the increased occurrence of summer hot temperature extremes over several midlatitude regions (Horton et al., 2015; Jézéquel et al., 2018, 2020). We first apply dynamical adjustment at daily time scale to all summer days of the 1979–2018 period for both maximum (*TX*) and minimum temperature (*TN*). We then identify maximum and minimum temperature extreme hot days for every summer of the 1979–2018 period and estimate changes in temperature extremes as well as the role of atmospheric circulation in these changes based on the dynamical adjustment results. We focus on two specific
regions, loosely defined as western Europe (from 35° N to 65° N and 15° W to 25° E) and western Asia (from 35° N to 65° N and 25° E to 60° E). Based on a trend analysis of atmospheric circulation patterns derived from a self-organizing map clustering approach, Horton et al. (2015) have attributed a fraction of the increase in the occurrence of summer hot extreme days for these two regions to an enhanced occurrence frequency and–or persistence (and/or duration) of anticyclonic circulation patterns during the 1979–2013 period. Here we assess whether a different but complementary approach can be
used to investigate whether atmospheric circulation changes have contributed to changes in maximum and minimum temperature summer extremes over a slightly extended period (1979–2018). We restrict our analysis to hot (*TX* maxima) and cold (*TN* maxima) summer extremes.

The paper is organized as follows. Section 2 describes the observational and reanalyses datasets and the methodological
aspects of the dynamical adjustment approach. Section 3 presents the results for the two illustrative extreme events and a comparison with other approaches based on published results. Based on the dynamical adjustment approach, section 4 then investigates the possible contribution of changes in atmospheric circulation patterns to the recent (1979–2018) increase in summer hot and cold extremes over western Europe and western Asia. Finally, section 5 gives a short summary and possible directions for future work.

**2 Material and Methods**

**2.1 Observational and reanalyses datasets**

**2.1.1 Mean sea level pressure data from reanalyses**

We mainly use daily mean sea level pressure (*SLP*) from the from the 2° × 2° Twentieth Century Reanalysis version 3 (20CR_V3, Slivinski et al., 2019) from 1836 to 2015 to characterize atmospheric circulation patterns and their link with
temperature extremes. The data are extended through 2018 with ERA-Interim (ERAI ; Dee et al. 2011) by adding daily ERAI anomalies to the daily 20CR_V3 climatology (based on the 1979–2015 period which is the common period between

ERAI and 20CR_V3). We also use daily *SLP* data from 20CR version2c (20CR_V2C, Compo et al., 2011), also extended through 2018 with ERAI. For both 20CR_V3 and 20CR_V2C, we only use daily *SLP* data from 1900 to 2018 due to the sparsity of the observational record in the 19th century. Finally, we also make use of the NCAR/NCEP-R1 (Kalnay et al.,
1996) on the shorter period (1948–2018) to further assess the sensitivity of the dynamical adjustment results to the choice of atmospheric reanalysis for the sea level pressure field.

### 2.1.2 Temperature datasets

The Berkeley Earth temperature (BERK) daily datasets are experimental products (http://berkeleyearth.lbl.gov/auto/Global/Gridded/Gridded_Daily_README.*Tx*t) and are available from 1880-01-01 to
2018-12-31. The BERK datasets are homogenized daily temperature fields built as a refinement upon their monthly temperature datasets (Muller et al. 2013; Rohde et al. 2013) and using similar techniques. The gridded data are provided on a regular latitude-longitude grid at 1-degree resolution. We only consider temperature data over the 1900–2018 period to match the period chosen for mean sea level pressure.

The EOBS daily land surface air temperature gridded datasets are also used over western Europe. The homogeneous EOBS
dataset (version19.0eHOM) is available from 1950-01-01 to 2018-11-30 (Cornes et al. 2018; Squintu et al. 2019). The raw station data are first homogenized using a quantile matching technique (Squintu et al. 2019). The gridded temperature data is provided on a regular latitude-longitude grid at 0.25-degree resolution (https://www.ecad.eu/download/ensembles/downloadversion19.0eHOM.php#datafiles). The data is provided for a geographical domain from 25° N to 71.5° N and from 25° W to 45° E.
The HadGHCND global product has been created based on daily station observations from the Global Historical Climatology Network-Daily database (Caesar et al., 2006). This consists of over 27,000 stations with temperature observations, though the temporal and spatial coverage of the record is very variable. Quality control has been carried out to indicate potentially spurious values. The temperature data is provided as anomalies relative to the 1961–1990 reference period. The HadGHCND dataset spans the years 1950 to 2014 and is available on a 2.5° latitude by 3.75° longitude grid.

The BERK and EOBS datasets are used as our reference temperature datasets.  HadGHCND as well as the NCEP reanalyses are also used to complement the observational uncertainty analysis for temperature. Unless explicitly mentioned, all *TX* and *TN* anomalies are calculated relative to the 1981–2010 reference period.

### 2.2 Dynamical adjustment based on constructed analogues

The dynamical adjustment used in this study is a straightforward adaptation to daily time scales of the method introduced in Deser et al. (2016). The main objective of dynamical adjustment is to derive an estimate of the component of any physical variable variability due solely to atmospheric circulation changes. In agreement with many previous studies, we assume that robust forced circulation changes over the North Atlantic European domain are not currently detectable due to a small signal-

to-noise ratio. Consequently, observed circulation changes are considered as being an integral part of climate internal variability. In the following and for the sake of concision, we refer to any variable changes due to atmospheric circulation as to the dynamic component (instead of the internal dynamic component). Here, $SLP$ is used to represent atmospheric circulation changes and we use $TX$ as our physical variable in the method description. Consequently, dynamical adjustment leads to the decomposition of any daily $TX$ anomaly between a $TX$ dynamic component and a residual (loosely described as the "thermodynamic residual" or simply the thermodynamic component). Note that the thermodynamic component may include both forced and internal contributions.

We now briefly summarize the dynamical adjustment algorithm. We first define a geographical domain for $SLP$ with the constraint that dynamically-adjusted $TX$ values are only meaningful in a region enclosed within the $SLP$ domain and having a smaller longitudinal and latitudinal extent than the $SLP$ one. The geographical boundaries of the $SLP$ domains are 25°N–90°N, 60°W–100°E and 25°N–90°N, 20°W–80°E for the 2009–2010 cold European winter and the 2010 Russian heat wave, respectively. For any day $d_i$ of the extreme event, we search for the closest $N_a$ daily $SLP$ analogues in all years (but the one of the extreme event occurrence) within a time window of $\pm N$ days centered on $d_i$ ($N$ being typically $\sim 15$ days). The $SLP$ analogues are ranked according to the Teweless-Wobus skill score. The score measures the similarity between the $SLP$ horizontal gradients (i.e geostrophic winds). We then randomly subsample (without replacement) $N_s$ of the $N_a$ $SLP$ analogues and compute their best linear fit (see Appendix of Deser et al., 2016 for details) to the target $SLP$ field (that of day $d_i$). The dynamically-reconstructed $TX$ is then defined as the corresponding linear combination of daily $TX$ anomalies associated with the $N_s$ $SLP$ analogues. Next, we repeat this random subsampling procedure $N_r$ times. Finally, we average the $N_r$ optimal sets of reconstructed daily $SLP$ analogues and associated $TX$ to obtain the dynamic component, defined as the "best estimate" of the circulation-induced component of maximum temperature anomaly for the day $d_i$. This sequence of steps is finally repeated for all days of the extreme event under consideration. Uncertainty estimates can be derived with a simple bootstrap procedure applied to the set of $N_r$ estimates of the $TX$ dynamic component (O'Reilly et al., 2017). We randomly draw (with replacement) $N_r$ estimates 1000 times to produce a distribution that can then be used to derive a 95 % confidence interval. The uncertainty analysis can be applied for any single day of the extreme event (Figs. 2a and 4a) or to the time-averaged event magnitude (Figs. 1d and 3d). In the latter case, we randomly draw $N_r$ estimates (with replacement) among the $N_r$ ones for every day of the event, take the average of the $N_r$ estimates and repeat the process 100 times. We end up with 100 estimates of the dynamic component for each day of the event. We then randomly select one estimate for each day of the event, take the time average, and repeat the process 1000 times to get the final distribution.

All results shown below are based on the following parameter values: $N_a = 400$, $N_s = 200$ and $N_r = 100$ (see parameter sensitivity tests in Appendix A). We have also checked that the selected analogues span the whole period evenly and do not preferentially arise from a specific multidecadal period such as the data-rich recent one (see Appendix B). As we are interested in separating the $TX$ dynamic component from any forced thermodynamic residual (due for instance to changes in

the external forcing), we need to remove a local estimate of the forced *TX* component before applying dynamical adjustment. In a sense, the *TX* dynamic component (*DYN$_{CF}$* thereafter) represents the effect of atmospheric circulation on the *TX* anomaly in the counterfactual world (the world with no human influence on climate). As one of the objectives of this approach is to rely exclusively on observations (and/or reanalyses), we apply a Loess-based smoother (see section 2.3) to *TX* daily observations to remove the low-frequency trend (for all grid-points) that we hypothesize to be primarily due to external forcing (Hawkins et al. 2020; Section 2.3).

Our physical interpretation of the *TX* dynamic component is that it represents the "mean" contribution of the atmospheric circulation pattern, including both direct (advection) and indirect (e.g local feedbacks) effects, in the counterfactual world. Here, the use of "mean" is simply associated with an average over multiple linear combinations of *TX* anomalies arising from a large of number of days having different ocean and/or land surface conditions.

We then interpret the residual component (*RES$_{TOT}$*) as being the sum of three contributions. The first one (*RES$_{TRD}$*) is the externally-forced *TX* component that has been removed before applying dynamical adjustment. The residual component also includes any *TX* changes due to a local or remote contribution associated with internal variability (*RES$_{INT}$*). For example, the local contribution includes local processes such as those associated with land surface feedbacks linked to soil moisture or snow cover anomalies. The remote contribution includes any *TX* change related to thermal advection changes due to mean flow advection of anomalous zonal and meridional *TX* gradients caused by internal variability (for example due to anomalous oceanic air masses). The last contribution (*RES$_{FRC}$*) includes thermal horizontal advection changes related to externally-forced changes in zonal and meridional *TX* gradients as well as forced changes in other factors such as radiative processes and vertical advection anomalies (Pfahl and Wernli, 2012; Quinting and Reeder, 2017). The estimation of *RES$_{INT}$* and *RES$_{FRC}$* can be obtained by running the dynamical adjustment twice: firstly, with the *TX* forced response removed (as previously described) and secondly with the observed raw *TX*. The *RES$_{FRC}$* contribution can then be estimated by subtracting the former *TX* dynamic component (from the counterfactual world) from the latter one (from the factual world). Finally, *RES$_{INT}$* can be estimated as:

$$RES_{INT} = RES_{TOT} - RES_{TRD} - RES_{FRC} \qquad (1)$$

The final decomposition of any daily *TX* anomaly (*TX$_A$*) can then be written as:

$$TX_A = DYN_{CF} + RES_{INT} + RES_{TRD} + RES_{FRC} \qquad (2)$$

With the objective to compare with model-based studies (see section 3.2) and assuming that the contribution of forced changes in radiative processes is not the dominant factor, it is also useful to define an upper bound of the "total" dynamic contribution $DYN_{TOT}$ given by:

$$DYN_{TOT} = DYN_{CF} + RES_{FRC} \qquad (3)$$

**2.3 Estimation of the forced response**

We assume that the temperature forced response to external forcing can be simply estimated with a low-frequency trend estimated over the 1900–2018 period. The latter is estimated with a Loess smoother (Cleveland et al., 1990) as implemented in the NCSTAT package (https://terray.locean-ipsl.upmc.fr/ncstat/index.html). We choose a smoother length of 45 years and we apply a light (~ 2 years) additional smoothing of the trend before estimating the residual. Iterations are carried out until

convergence of the trend, which is reached when maximum changes in individual trend fits are less than 1 % of the trend's range after the previous iteration. We detrend the daily $TX$ and $TN$ datasets separately for each month before applying the dynamical adjustment procedure and estimating the dynamic component.

**3 Results for individual extreme events**

As our illustrative examples, we choose two seasonally contrasted extreme events that have been widely documented in the

240 literature: the cold European winter of 2009–2010 and the 2010 Russian summer heatwave. For the Russian heatwave, we follow previous studies by focusing on the July $15^{th}$ – August $14^{th}$ period. For the cold European winter, we choose a seventeen-day period between December $28^{th}$ 2009 and January $13^{th}$ 2010 that is associated with record-breaking temperatures in many midlatitude land masses of the Northern Hemisphere (Wang et al., 2010). We restrict our analysis to the $TX$ variable. For each illustrative example, we first describe the synoptic circulation and associated $TX$ anomalies during

the event before showing the dynamical adjustment results averaged over all event days. We then briefly discuss the chronology of the event and the evolution of the $TX$ dynamic component. We use 20CR_V3, BERK and EOBS as primary datasets for our dynamical adjustment analysis and figures in the main text. Specifically, we use EOBS for the 2010 winter event and BERK for the summer one (note that the EOBS geographical domain does not cover Russia). Results based on the other $TX$ and $SLP$ datasets are shown in Tables 1 and 2.

**3.1 The 2009-2010 European winter cold spell**

Winter 2010 is characterized by an extreme negative phase of the North Atlantic Oscillation (NAO) (the classical NAO index reaches a value of 3 standard deviation below average, see Cattiaux et al., 2010 and Osborn 2011). In the eastern Atlantic the winter (December–February) mean eddy-driven jet was displaced southward by almost 10 degrees compared with its climatological position and maintained south by diabatic heating feedbacks (Woollings et al., 2016). Averaged $SLP$

anomalies during the extreme event period (December 28th 2009 – January 13th 2010) display a dipole with large positive anomalies over the north-western Atlantic and negative ones over the central eastern Atlantic, in agreement with a jet stream axis located over Northern Africa (Fig. 1a). Importantly, the reconstructed *SLP* pattern is almost identical to the original observed *SLP* pattern (Fig. 1b). This anomalous *SLP* pattern strongly projects onto the negative NAO pattern. Negative NAO phases are known to lead to cold temperature over western and northern Europe (Hurrell 1995). The spatial pattern of the *TX* anomaly during the cold spell displays an elongated cold *TX* anomaly over the United Kingdom and northern Europe contrasting with warm *TX* anomalies in Northern Africa and the Middle East (Fig. 1a). The magnitude of the mean *TX* anomaly for the cold spell event – regionally averaged over the European domain (see box in Fig. 1c) – is -2.04 °C based on EOBS. As expected, the dynamic component contribution to the *TX* anomaly is negative and has a larger magnitude than the total (with a mean and 95 % confidence interval of -2.76 °C and [-2.95 °C, -2.57 °C]). In particular, the dynamic component displays very cold (~ -5 °C) *TX* anomalies over northeastern Europe (Fig. 1b). The total residual contribution is positive (Fig. 1c) and has a smaller amplitude (0.72 °C) than the dynamic component due to the opposite sign of the internal residual contribution (-0.31 °C, Fig. 1d) and the two forced contributions, the long-term trend ($RES_{TRD}$: 0.44 °C, Fig. 1e) and the residual forced component ($RES_{FRC}$: 0.59 °C, Fig. 1f). The total *TX* forced contribution (defined as the sum of $RES_{TRD}$ and $RES_{FRC}$) has a significant positive contribution (1.03 °C) and shows increased warming in northern Europe (Fig. 1e-f).

It is noteworthy that the internal residual contribution displays coherent large-scale patterns with grid-point values that are outside of the uncertainty range of the dynamic component (Fig. 1d), suggesting that its salient regional features are related to other factors than dynamical ones. The $RES_{INT}$ pattern exhibits cold *TX* anomalies along the coasts of western Europe, perhaps linked to cold and persistent – present in both December 2009 and January 2010 – North Atlantic SST anomalies (Buchan et al., 2014). These SST anomalies may have been the surface signature of a reduced northward ocean heat transport related to a strong decrease of the Atlantic meridional overturning circulation in 2009 (McCarthy et al., 2012; Sonnewald et al., 2013). We speculate that the amplitude of these ocean-induced cold SST anomalies has been further enhanced in late winter due to the ocean integration of the recurrent and persistent negative NAO atmospheric forcing.

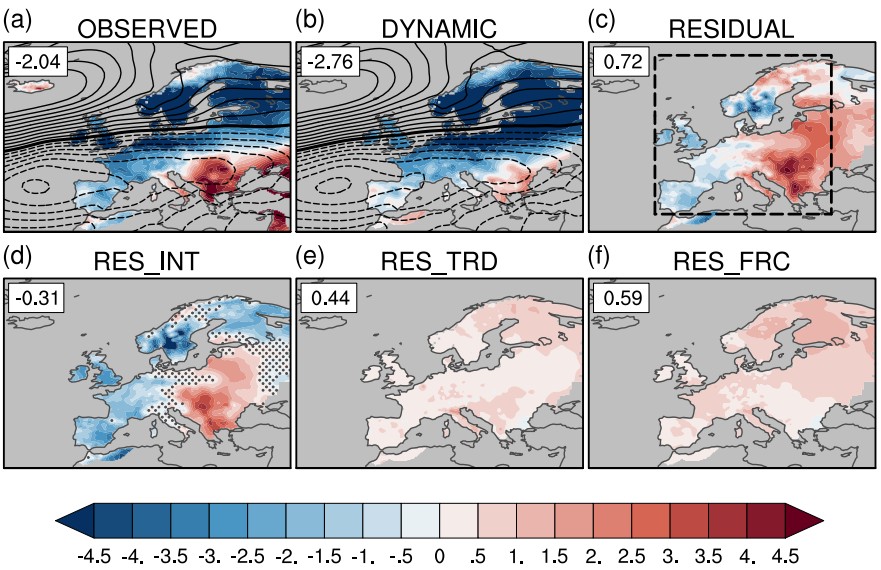

**Figure 1:** EOBS daily maximum temperature (° C, shading) and 20CR_V3 sea level pressure (hPa, black line contours with contour interval of 1 hPa) anomalies averaged over the European cold spell period (December 28th 2009 – January 13th 2010): **(a)** total *TX* anomaly and observed *SLP* anomaly, **(b)** *TX* dynamic component contribution and reconstructed *SLP* anomaly, **(c)** *TX* total residual contribution, **(d)** *TX* internal residual contribution **(e)** *TX* long-term trend residual contribution, **(f)** *TX* residual contribution from forced changes in other factors. Numbers in the upper right corner indicate the weighted average *TX* anomaly over the region delimited by the black dashed box in **(c)**. In **(a)** and **(b)**, the black thick contour line indicates the zero *SLP* anomaly and dashed contour lines indicate negative *SLP* anomalies**.** In (d), stippling indicates grid-points where $RES_{INT}$ values are within the uncertainty range of the dynamical component given by the 95 % confidence interval estimated by the bootstrap method given in section 2.2.

Further inland, the internal residual contribution shows warm *TX* anomalies with maximum values in south-eastern and central Europe (Fig. 1d). We speculate that the Black Sea and Levantine sub-basin warm SST anomalies observed in 2009 and 2010 (in line with the eastern Mediterranean SST warming trend over 1982–2012, Shaltout and Omstedt, 2014) may have contributed to the internal residual warm *TX* anomalies in south-eastern Europe.

These results confirm that the long-term warming (mostly attributed to human influence) has mitigated the extreme character of the 2009-2010 early winter cold spell as initially suggested by Cattiaux et al. (2010). In the counterfactual world, the winter cold spell would have been 1.03 °C (Fig. 1e-f) colder than the observed one (-3.07 °C instead of -2.04 °C). Keeping 20CR_V3 for *SLP*, additional results based on BERK and HadGHCND for *TX* lead to quasi-similar amplitudes for the total anomaly and dynamic component (Table 1). The use of NCEP data for *TX* underestimates the amplitude of the two-week cold spell by 35 %. Using different datasets for *SLP* (20CR_V2C and NCEP) while keeping EOBS for *TX* leads to slightly larger values of the dynamic component.

We now illustrate how the dynamical adjustment approach can be used to track the daily evolution of the contribution due to atmospheric circulation changes. The chronology of the early 2010 winter cold spell shows two cold minima, the first one in mid-December and the second one, more persistent, two weeks later (Fig. 2a). The dynamic component is by far the main

contributor to the *TX* anomaly magnitude daily and weekly variability (as suggested by the similarity of the two timeseries in

Fig. 2a). In particular, the two cold minima observed during December 2009 and January 2010 are associated with an eastward extension of the anticyclonic *SLP* anomalies centered around Iceland that favors the advection of Arctic air masses towards northern Europe (Fig. 2b). While observed daily NAO index values (https://www.cpc.ncep.noaa.gov/products/precip/CWlink/pna/nao.shtml) are all negative during the period shown in Fig. 2a (albeit with different amplitudes) illustrating the persistence of the low-frequency large-scale atmospheric flow, the

contribution of the dynamic component to *TX* anomaly exhibit significant daily variability due to the high-frequency part of the flow. For instance, the circulation-induced and region-averaged *TX* anomaly on December 31st is positive, which contrasts with the cold minimum observed a few days later and associated with a marked negative NAO phase (Fig. 2).

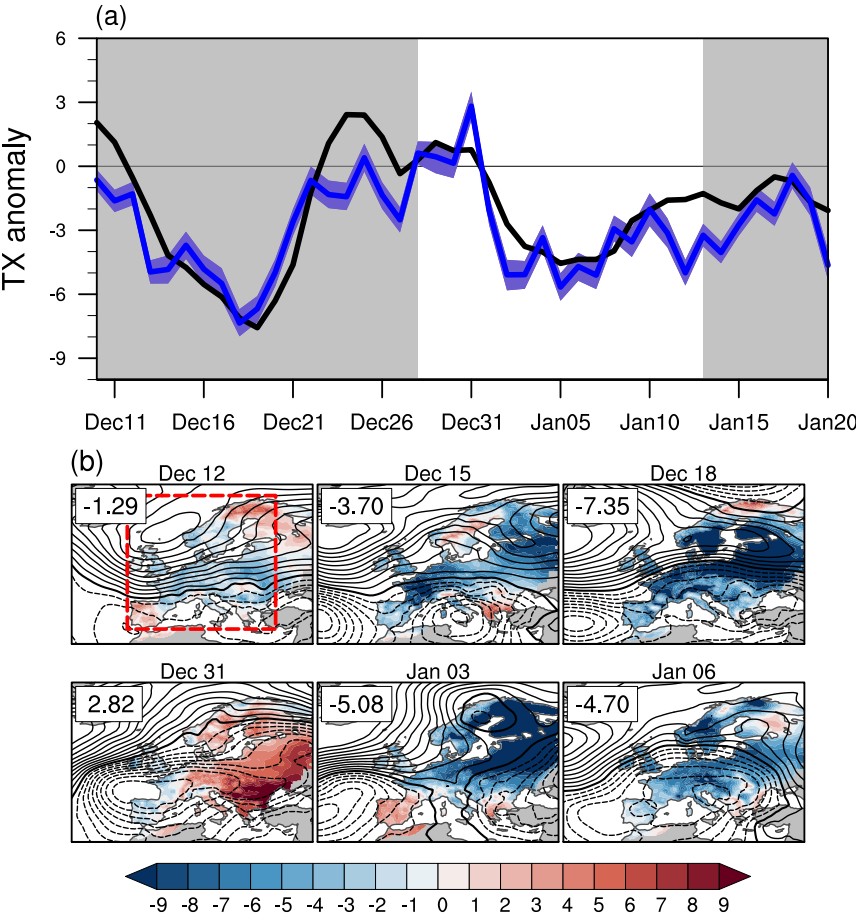

**Figure 2: (a)** time evolution of EOBS daily maximum temperature (° C) anomaly averaged over the European domain (box with red
dashed line in **(b)**). The covered period is from December 10, 2009, until January 20, 2010. The thick black line represents the total *TX* anomaly. The thick blue line shows the contribution of the dynamic component and the blue shading indicates the 95 % confidence interval of the reconstruction based on bootstrapping (see section 2.2). The chosen period for the two-week cold spell is defined by the white background. **(b)** Daily maps of *SLP* anomaly (hPa, black line contours with contour interval of 1 hPa) and of the dynamic component contribution (°C, shading) to the total *TX* anomaly. The black thick contour line indicates the zero *SLP* anomaly and dashed

contour lines indicate negative *SLP* anomalies. Numbers in the upper left corner indicate the region-averaged contribution of the dynamic component to the total *TX* anomaly.

**Table 1:** Total *TX* anomaly (°C, in bold) and dynamic component contribution (°C and as a fraction of the total anomaly in percent) averaged over Europe (36° N–72° N; 10° W–30° E, black box in Fig. 1c) during the 2009–2010 winter cold spell. The reference period is 1981–2010.

| SLP ╲ TX | 20CR_V3 | 20CR_V2C | NCEP |
|---|---|---|---|
| NCEP | **-1.32** – -1.96 (148 %) | **-1.32** – -2.78 (210 %) | **-1.32** – -2.40 (182 %) |
| HadGHCND | **-2.20** – -2.83 (129 %) | **-2.20** – -3.52 (160 %) | **-2.20** – -3.04 (138 %) |
| EOBS | **-2.04** – -2.76 (135 %) | **-2.04** – -3.59 (176 %) | **-2.04** – -3.36 (165 %) |
| BERK | **-1.95** – -2.63 (135 %) | **-1.95** – -3.01 (154 %) | **-1.95** – -3.04 (156 %) |

### 3.2 The 2010 Russian summer heatwave

Summer 2010 is characterized by persistent quasi-stationary anticyclonic circulation anomalies over western Russia (Dole et al., 2019; Barriopedro et al., 2010). The persistence of the long-lasting blocking high has been linked to a transition between ENSO warm and cold phases and the resulting changes in quasi-stationary wave anomaly and transient eddies (Schneidereit et al., 2012; Drouard and Woollings, 2018). These blocking circulation patterns are often associated with surface temperature warm anomalies due to enhanced subsidence and adiabatic compression, reduced cloudiness allowing a greater fraction of solar radiation to reach the surface and horizontal advection of warmer air masses from regions located to the south of the blocks. A late-winter to spring precipitation deficit over western Russia has also likely contributed to the abnormally warm summer maximum temperature anomalies (with a magnitude on the order of ~ 9–10 °C when regionally-averaged over the heat wave period, see Wehrli et al., 2019 and Fig. 3 and Table 2) through the concurrent summer drought and associated land-surface feedbacks related to depleted soil moisture content (Miralles et al., 2014). Based on atmospheric model nudging experiments, Wehrli et al. (2019) have estimated dynamic (related to atmospheric circulation changes) changes and other contributions to the Russian heatwave. They suggest that the largest contribution to the Russian heatwave *TX* anomaly can be attributed to atmospheric circulation (range 54–63 %) with a substantial albeit smaller contribution (27–36 %) from antecedent soil moisture conditions (the remaining 10 % being due to the contribution of the response to external forcing, named greenhouse gas forcing in their paper).

Figure 3 shows our estimates of the different contributions based on the dynamical adjustment approach and the BERK dataset. The event total *TX* anomaly for the western Russia region (see box in Fig. 3b) is 9.06 °C and is located southwest of the blocking high maximum (with a *SLP* magnitude of 9 hPa, Fig. 3a). As suggested by previous studies (Dole et al., 2011;

Wehrli et al., 2019), we find that the total *TX* anomaly is dominated by the total internal contribution ($DYN_{CF}$ [3.99 °C] + $RES_{INT}$ [3.59 °C]) with the total forced contribution ($RES_{TRD}$ [1.03 °C] + $RES_{FRC}$ [0.45 °C]) being only 16 % of the total *TX* anomaly (Fig. 3). The magnitude of the dynamic component is 3.99°C with an uncertainty range (given by the 95 % confidence interval) of [3.88 °C, 4.1 °C]. The magnitude of the total dynamic contribution is 4.44 °C ($DYN_{TOT} = DYN_{CF}$

[3.99 °C] + $RES_{FRC}$ [0.45 °C]; ~ 49 % of the event total anomaly). The other and smaller contributions are the *TX* long-term trend residual contribution $RES_{TRD}$ (1.03 °C, ~ 11 % of the total) and the internal residual $RES_{INT}$ (3.59 °C, ~ 40 % of the total). Using HADGHCND or NCEP data for *TX* leads to very similar results in term of the percentage of the dynamic contribution (Table 2). The latter is slightly lower when *SLP* from the 20CR_V2C and NCEP datasets is used for dynamical adjustment while keeping BERK for *TX*.

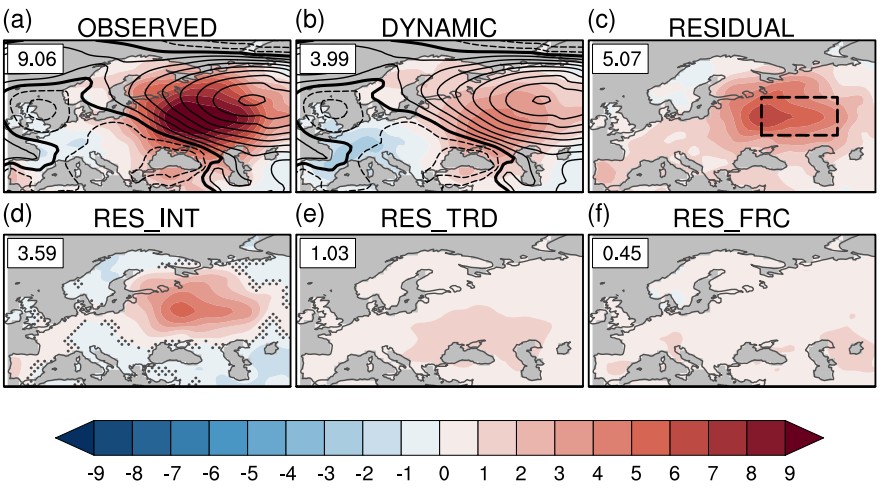


**Figure 3:** BERK daily maximum temperature (*TX*, ° C, shading) and 20CR_V3 sea level pressure (*SLP*, hPa, black line contours with contour interval of 1 hPa) anomalies averaged over the Russian heatwave period (July 15[th] – August 14[th], 2010): **(a)** total *TX* anomaly and observed *SLP* anomaly, **(b)** *TX* dynamic component contribution and reconstructed *SLP* anomaly, **(c)** *TX* total residual contribution, **(d)** *TX* internal residual contribution **(e)** *TX* forced residual contribution, **(f)** *TX* thermal advection residual contribution. Numbers in the upper

right corner indicate the weighted average *TX* anomaly over the region delimited by the black dashed box in **(c)**. In **(a)** and **(b)**, the black thick contour line indicates the zero *SLP* anomaly and dashed contour lines indicate negative *SLP* anomalies. In (d), stippling indicates grid-points where $RES_{INT}$ values are within the uncertainty range of the dynamic component given by the 95% confidence interval estimated by the bootstrap method given in section 2.2.

Therefore, we find that the total dynamic component yields the dominant contribution to the Russian heatwave *TX* anomaly,

in agreement with the model study of Wehrli et al. (2019). We also note that our best estimate for the total dynamic component contribution (49 % of the total *TX* anomaly) is slightly lower than their minimum estimate (see discussion below). As suggested by many previous studies, it is very likely that anomalous soil moisture is an important contributor to the substantial magnitude of the internal residual contribution (note however that the dynamical adjustment approach cannot be used directly to infer the soil moisture influence, see also the discussion below). The magnitude of the *TX* long-term trend

residual contribution is in reasonable agreement with the model-based estimate (1.2 °C) of Wehrli et al. (2019).

Differences with results from the latter study regarding the magnitude of the dynamic contribution (and consequently, of the thermodynamic residual magnitude) can be due to multiple factors. First, using the same baseline (1982–2008) as Wehrli et al. (2019) doesn't significantly change the contribution of the dynamic component (46 % instead of 49 %). Second, while our methodology only relies on observations and reanalysis, their approach relies upon both "all-but-one" and "only-one"

modelling frameworks based on simulated differences between SST-forced historical atmospheric experiments with and without circulation and/or soil moisture nudging and a control simulation without any nudging. Possible caveats of this modelling approach include the lack of validity of the assumption that the different factors are additive, the lack of interaction between the ocean and the atmosphere and the fact that different soil moisture climatology between simulations with and without soil moisture nudging can lead to a different response to the same soil moisture anomaly (for a detailed

discussion, see Wehrli et al., 2019). Possible sources of uncertainty of the dynamical adjustment results include observational uncertainty and the presence of "noise" in the internal thermodynamic residual resulting from dynamic contributions not accounted for by the constructed analogue technique (due to both inadequate sampling as well as methodological uncertainty). Table 2 shows that the magnitude of the dynamic component for different observational and reanalysis products is always less than 50 % of the total *TX* anomaly. This suggests that differences in magnitude of the

dynamic component discussed above are unlikely to be fully explained by observational uncertainty in the dynamical adjustment method.

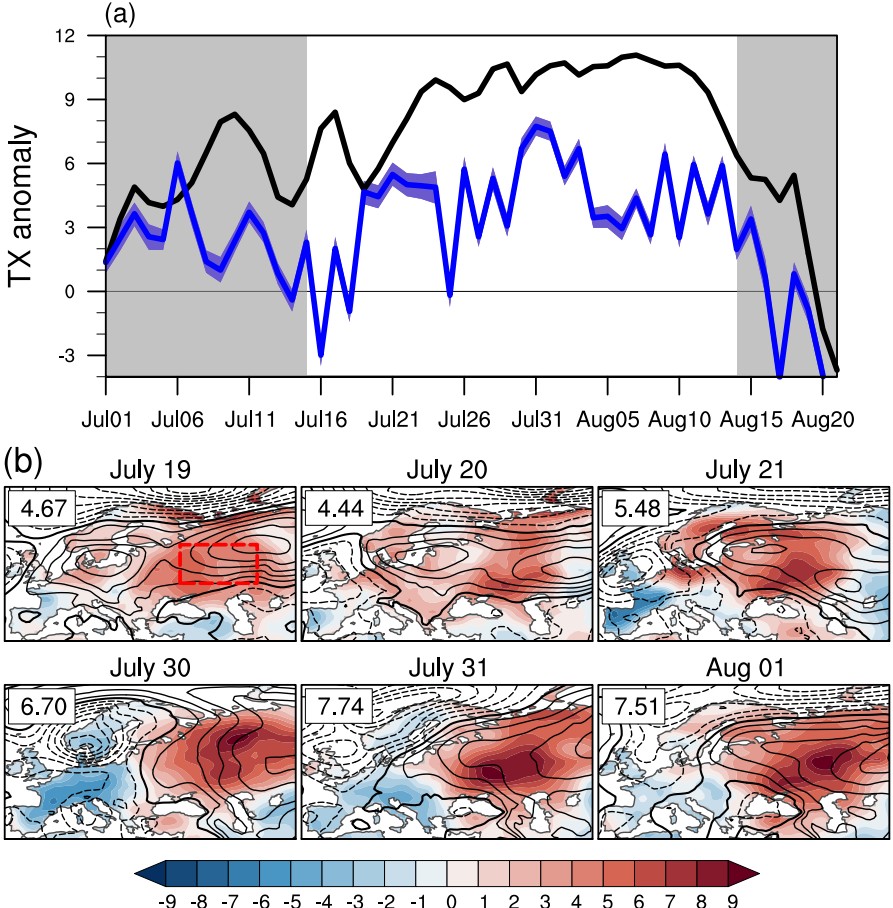

**Figure 4: (a)** time evolution of BERK daily maximum temperature (° C) anomaly averaged over the Western Asia domain (box with red dashed line in **(b)**). The covered period is from July 1$^{st}$, 2010, until August 20$^{th}$, 2010. The thick black line represents the total *TX* anomaly. The thick blue line shows the contribution of the dynamic component and the blue shading indicates the 95 % confidence interval of the reconstruction based on bootstrapping (see section 2.2). The chosen period for the Russian heatwave is defined by the white background. **(b)** Daily maps of *SLP* anomaly (hPa, black line contours with contour interval of 1 hPa) and of the dynamic component contribution (°C, shading) to the total *TX* anomaly. The black thick contour line indicates the zero *SLP* anomaly and dashed contour lines indicate negative *SLP* anomalies. Numbers in the upper left corner indicate the region-averaged contribution of the dynamic component to the total *TX* anomaly.

The chronology of the Russian heatwave suggests that the contribution of the dynamic component to the total *TX* anomaly varies at daily timescale (Fig. 4a). As expected, the dynamic component seems to play a key role in the initiation and termination of the heatwave. In particular, the negative contribution from the dynamic component after August 15$^{th}$ is leading by a couple of days the decline of the extreme heat. During the heatwave, two multi-day periods (July 19–23 and July 30 – August 1) show persistent and high values of the *TX* dynamic component (Fig. 4b). The first one (the less extreme) is associated with a zonally-extended anticyclonic anomaly from the European Coasts to western Russia. The largest contributions of the dynamic component appear to be associated with a strong blocking High center situated eastward of the location of maximum *TX* anomaly.

**Table 2:** Total *TX* anomaly (°C, in bold) and dynamic component contribution (°C and as a fraction of the total anomaly in percent) averaged over western Russia (50° N–60° N; 35° E–55° E, black box in Fig. 3c) during the 2010 summer Russian heat wave. The reference period is 1981–2010.

| SLP / TX | 20CR_V3 | 20CR_V2C | NCEP |
|---|---|---|---|
| NCEP | **10.16** – 3.85 (38 %) | **10.16** – 3.67 (36 %) | **10.16** – 3.05 (30 %) |
| HadGHCND | **9.16** – 3.75 (41 %) | **9.16** – 3.51 (38 %) | **9.16** – 3.31 (36 %) |
| BERK | **9.06** – 3.99 (44 %) | **9.06** – 3.07 (34 %) | **9.06** – 3.2 (35 %) |

## 4 Atmospheric circulation contribution to recent changes in summer temperature hot and cold extremes

We now use dynamical adjustment to assess the possible changes in circulation-related temperature anomalies and their contribution to summer temperature hot extreme changes during the 1979–2018 period. We select two Northern Hemisphere midlatitude regions, western Asia (*TX*, *TN* domain: 35° N–65° N; 25° E–60° E*; SLP* domain: 25° N–80° N; 10° E–65° E) and western Europe (*TX*, *TN* domain: 15° E–25° W; 35° N–70° N; *SLP* domain: 30° E–45° W; 25° N–80° N), and apply the dynamical adjustment separately to each of them. The focus is on the contribution of the dynamic component to changes in the summer warmest day (*TX* maxima) and warmest night (*TN* maxima) temperature over these four decades. The initial step is to run the dynamical adjustment procedure for all summers during the 1979–2018 period and for both *TX* and *TN*. As with the two illustrative examples, we apply the dynamical adjustment procedure twice, with and without detrending, before applying dynamical adjustment. For each year and each grid-point, we then select the days with the largest *TX* and *TN* anomalies. In the following, we focus on the summer days and nights with the most extreme temperature: the days with the hottest maximum (*TXx*) and minimum (*TNx*) temperature. We then estimate changes in *TXx* and *TNx* during 1979–2018 by using the non-parametric Mann-Kendall test and Theil-Sen's estimator to calculate the trend. Based on dynamical adjustment results, we also quantify the contribution of both total dynamic (*DYN$_{TOT}$*, see Eq. (3)) and thermodynamic residual (*RES$_{INT}$* + *RES$_{TRD}$*) components to these changes.

### 4.1 Extreme maximum and minimum temperature trends in summer over western Asia and western Europe

We find warming trends for both *TXx* and *TNx* over large parts of the western Asia (WA) and western Europe (WE) regions (Figs. 5a, d and 6a, d). The warming of *TXx* over most of the WA region (often greater than 3 °C 40 yr$^{-1}$) is primarily due to the thermodynamic component with an additional and substantial contribution from the dynamic component north of the Black and Caspian Seas (Fig. 5b–c). Both thermodynamic and dynamic components contribute to the lack of statistical significance and small amplitude of the *TXx* trends found in the northeastern part of the WA region.

Warming trends in *TNx* are statistically significant only in a small region located south of the Black Sea while the eastern

part of the WA region exhibits significant cooling with contribution from both components (Fig. 5d–f). In addition, the lack of significant *TNx* trends over most of the WA region results from opposite effects from the thermodynamic (cooling) and dynamic (warming) component. This contrasts with the *TXx* case, supporting the existence of different processes governing the changes in *TXx* and *TNx*. For instance, clear-sky conditions often associated with subsiding motions near the central region of blocking patterns have opposite radiative effects on *TX* and *TN*: on one hand, they enhance daytime solar heating

leading to *TX* increase, and on the other hand, they also enhance nighttime longwave cooling leading to a *TN* decrease. Assuming that a substantial fraction of the dynamical component is related to the increased occurrence (and/or persistence) of blocking patterns during recent decades (Horton et al., 2015), one would expect a reduced amplitude of *TNx* changes compared with that of *TXx* changes in regions where these circulation changes have occurred. However, the eastern part of the WA region shows the opposite sign (cooling) with large amplitude for the *TNx* trend compared with that of *TXx*. Whether

this is a real signal or not is further discussed below in light of observational uncertainty. Finally, we have checked that omitting year 2010 has little influence on the raw *TXx* and *TNx* trend pattern and statistical significance, suggesting that the long-term signal is robust and not influenced by the exceptionally warm 2010 summer.

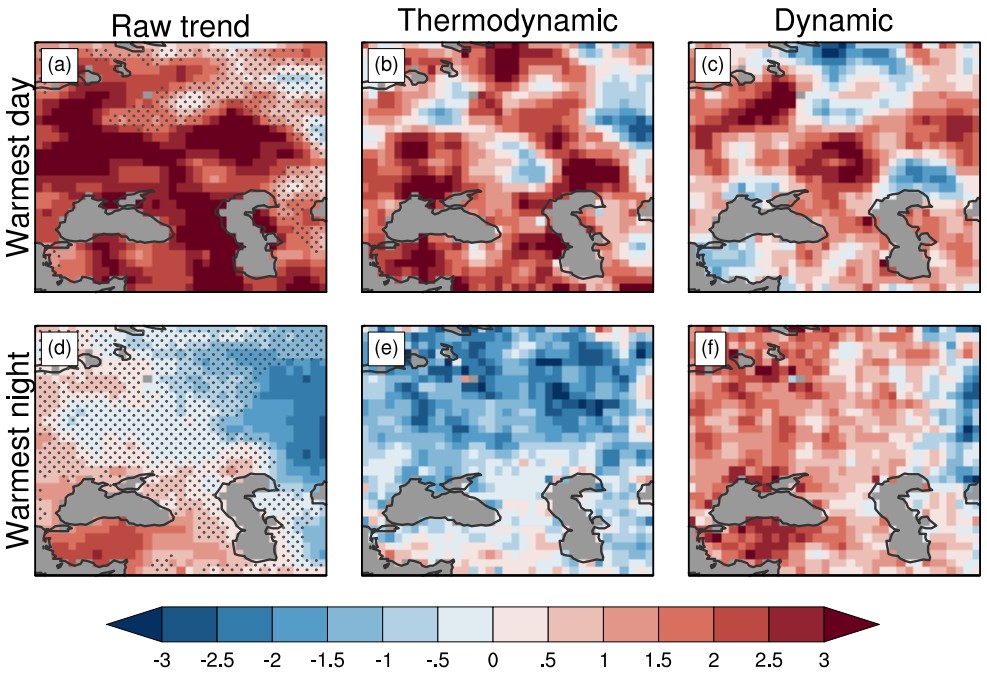

**Figure 5:** Summer temperature extremes: 1979–2018 trend maps for western Asia based on BERK (trend unit is ° C 40 yrs $^{-1}$, shading): (a-c) *TXx* raw trend, *TXx* thermodynamic and dynamic component trends. (d-f) *TNx* raw trend, *TNx* thermodynamic and dynamic component trends. The trend detection is estimated using the non-parametric Mann-Kendall test and the linear trend slope is computed based on the Theil-Sen estimator. In (a, d), stippling indicates locations where the raw trend is not significant at the 5 % level.

The dynamic contribution to the WA summer *TXx* trend magnitude can also be quantified with regard to year-to-year variability of the dynamic component that is quite similar (in both spatial pattern and amplitude) to the dynamic component daily variability during the Russian heatwave period. In WA regions with the largest trend magnitude (north of the Black and Caspian Seas), the 40-yr *TXx* changes are comparable with the summer *TXx* interannual standard deviation in term of localization and magnitude (not shown).

Regarding the WE region, the *TXx* trend map shows maximum warming (often greater than 3 °C 40 yr⁻¹) located over the central part of the domain and along the coasts of western Europe. This contrasts with most of southern Europe and Scandinavia where *TXx* trends are weaker and not statistically significant (Fig. 6a). Regions with significant *TXx* trends show different relative contributions from dynamic and thermodynamic components (Fig. 6a–c). Interestingly, the contribution of the dynamic component to the total trend is substantial over many locations, including northwestern Spain and France, northeastern Europe (east of the Baltic Sea) and northern Scandinavia (Fig. 6c). The *TNx* trend map shows widespread warming over western Europe and reduced amplitude compared with that of *TXx*, except for Italy and Greece, where the large trend values are mainly due to the thermodynamic component (Fig. 6d–f). Over the central Europe domain, both components contribute to the *TNx* warming with the contribution of the dynamic component being slightly dominant particularly over southern France, eastern Europe and Scandinavia (Fig. 6f).

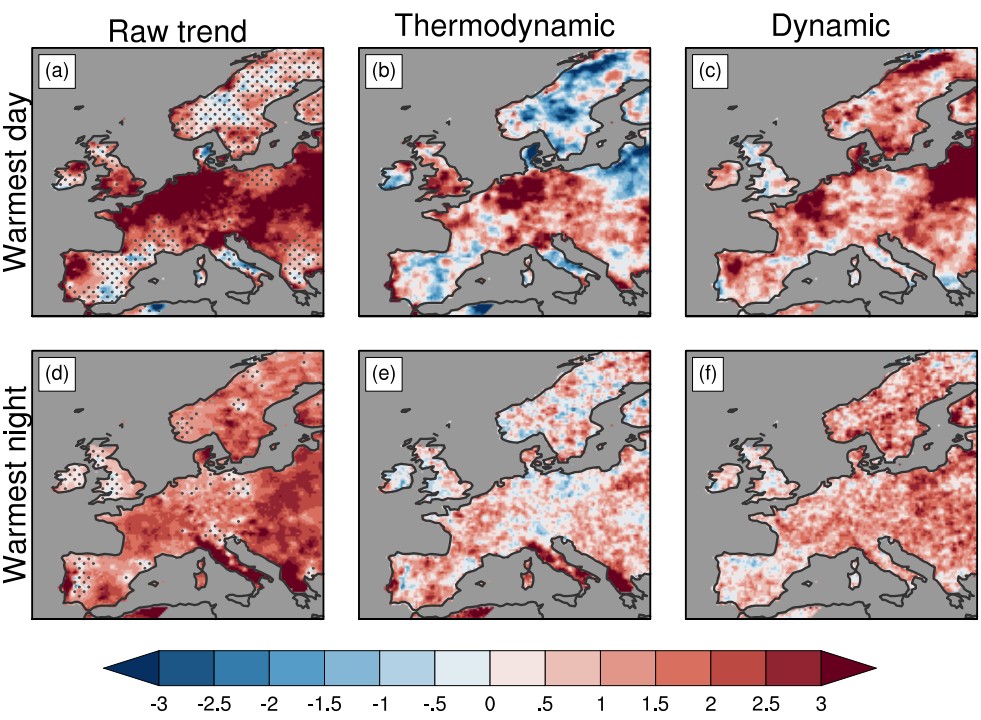

**Figure 6:** Summer temperature extremes: 1979–2018 trend maps for western Europe based on EOBS (trend unit is ° C 40 yrs ⁻¹, shading): **(a-c)** *TXx* raw trend, *TXx* thermodynamic and dynamic component trends. **(d-f)** *TNx* raw trend, *TNx* thermodynamic and dynamic

component trends. The trend detection is estimated using the non-parametric Mann-Kendall test and the linear trend slope is computed based on the Theil-Sen estimator. In **(a, d)**, stippling indicates locations where the trend is not significant at the 5 % level.

We now address the issue of observational uncertainty of the raw *TXx* and *TXn* trend analysis. We use the HadEX3 dataset (Dunn et al., 2020) to perform exactly the same *TXx* and *TNx* trend analysis as the one above with the BERK dataset. Figure 7a–b suggests that the main salient features of the trend patterns based on HadEX3 are reasonably similar to those derived from BERK for *TXx* and *TNx* over the WE region and *TXx* over WA. However, the substantial *TNx* cooling trend over the eastern part of the WA region seen with BERK (Fig. 5d) does not appear with HadEX3 (Fig. 7b). Instead, the HadEX3-based

*TNx* trend pattern exhibits a weak warming decreasing eastward. Looking further east shows that the cooling region exists in the HadEX3-based analysis but is shifted eastward compared with the BERK one (Appendix C, Fig. C1). We speculate that the difference in *TNx* trend patterns possibly arises from different sets of station data used in the two analyses, as well as differences in the optimal interpolation scheme such as different parameters of the distance-based correlation function.

4.2 Causal factors of the extreme temperature changes over the 1979–2018 period

We find that the dynamic component has substantially contributed to the increase in the summer *TX* and *TN* hottest extreme over parts of the WA and WE regions. The regions where the dynamic component is an important contributor to the *TX* warming trend broadly correspond to the ones suggested in Horton et al. (2015). This is especially noticeable for western Asia (see their figure 4i) where Horton et al. (2015) attributes a portion of the *TX* hottest extreme trend to an increase in blocking pattern occurrence (note that the trend spatial patterns shown in Figs. 5 and 6 and corresponding to the full period

1979–2018, are qualitatively similar to those for the reduced period 1979–2013 that is used in Horton et al., 2015, not shown). Increase in the occurrence of anticyclonic patterns mainly located in central Europe was also linked to the observed increase in the *TX* hottest extreme in eastern Europe (Fig. 3c, k of Horton et al. 2015). Our analysis confirms this result and suggests that the dynamic component has also been an important driver of the *TX* hottest extreme warming over several coastal areas of western Europe when considering the extended period up to 2018.

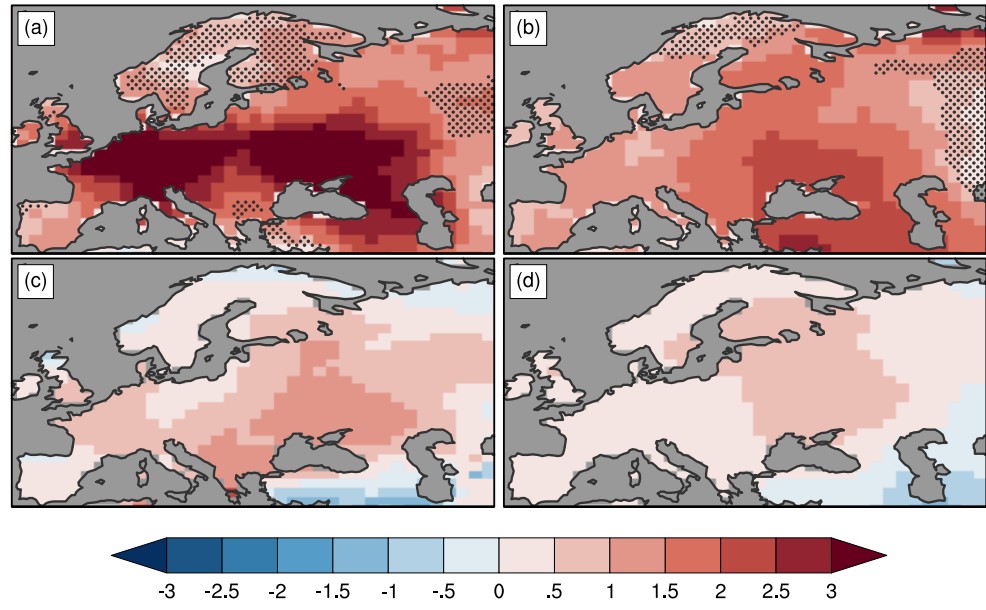

**Figure 7**: 1979–2018 Summer temperature trend patterns (units: °C 40 yrs-1) for both WE and WA regions based on the HadEX3 dataset: a) *TXx* and b) *TNx*. Summer (JJA) anomaly differences (units: °C) between warm and cold periods of AMV. The latter are defined as in O'Reilly et al. (2017): cold periods (1902–1925 and 1964–1993) and warm periods (1931–1960 and 1996–2012). The temperature data has been detrended before taking the difference between warm and cold periods. In **(a, b)**, stippling indicates locations where the trend is not significant at the 5 % level.

We now discuss some of the possible drivers of recent *TXx* and *TNx* changes. As the 1979–2018 period covers a transition between the negative (cold) and positive (warm) phase of the Atlantic Multidecadal Variability (AMV, Sutton and Dong 2012), the question arises as to whether the AMV phase shift has any influence on the temperature extreme trends. To estimate the AMV contribution to the total change in *TXx* and *TNx*, we have performed a simple composite analysis by calculating the temperature difference between warm and cold AMV periods (Fig. 7c–d). The AMV contribution to *TXx* changes is larger in the central part of the domain and varies from ~10 % over France to ~25 % west and north of the Black Sea. Regarding *TNx* changes, the AMV contribution is restricted to the region to the north of the Black Sea. South of the Black and Caspian Seas, the AMV contribution to *TXx* and *TNx* changes is seen to oppose the observed warming trends. Based on the summer mean temperature results from the observational study of O'Reilly and al. (2017), we speculate that the AMV shift may have contributed to both dynamic and thermodynamic *TXx* and *TNx* changes in western Europe and in the western part of western Asia (to about 45° E) over the 1979–2018 period.

In addition to the AMV influence, other factors have likely played a role on extreme temperature changes, in particular over the eastern part of the WA region where the AMV influence is weak. Interestingly, both HadEX3-based *TXx* and *TNx* trend patterns show a tripole pattern with two areas of accelerated warming over the Eastern European Plain and Central Siberia and a region of cooling over the Western Siberian Plain, which is located eastward of the eastern boundary of the WA region (Appendix C, Fig. C1). Sato and Nakamura (2019) have suggested that this tripole pattern (of daily mean temperature in

their study) is linked to the increased occurrence in the beginning of the 21[st] century of an unforced quasi-stationary wave train that has been anchored and amplified due to land-atmosphere interaction. Since the 1990s, there is evidence of

increasing precipitation, winter snow depth and snow cover extension over the Western Siberian Plain (Guo et al., 2019; Bulygina et al., 2009; Bulygina et al., 2011), leading to increasing snow melt in spring and soil moisture during summer as well as reduced sensible heat flux and negative temperature anomalies during summer (Sato and Nakamura 2019). We suggest that similar mechanisms may also be relevant for changes in maximum and minimum temperature extremes over the Western Siberian Plain.

## 520 5 Summary and discussion

The dynamical adjustment approach based on the constructed analogue method and extended at daily time scale has been used to assess the contribution of circulation-related temperature anomalies to temperature extreme events. Based on daily maximum temperature, two observed iconic extreme events have been selected to illustrate the potential of the approach: the early 2009–2010 winter European cold spell and the 2010 Russian heat wave. Dynamical adjustment results confirm the key

role and improve the quantification of the atmospheric circulation contribution (the dynamic component) to the two extreme events. The 2009–2010 winter European cold spell associated with an extreme negative NAO phase would have been significantly colder without human influence that mitigated the region-averaged amplitude of the cold extreme event by 33 %. Regarding the Russian heatwave, the contribution of the total dynamic component associated with persistent anticyclonic conditions during the 2010 summer is estimated to be close to 50 % of the observed maximum temperature anomaly.

Furthermore, we have used the dynamical adjustment approach to assess the possible contribution of atmospheric circulation to changes in summer extremes for two regions, western Asia and Europe, and during the 1979–2018 period. The dynamical adjustment results suggest that both dynamic and thermodynamic factors have contributed to observed changes in summer temperature extremes over the 1979–2018 period. We have focused on changes in the summer warmest day and night temperatures. Although thermodynamic influence has dominated the *TXx* changes in a large fraction of the western Asia

domain, the dynamic influence has been quite substantial north of the Black and Caspian Seas. Furthermore, the dynamic influence has been key in the *TNx* warming trend depicted south of the Black Sea. Regarding Europe, the influence of atmospheric circulation has been a major driver of both *TXx* and *TNx* warming trends that are seen in many regions, including the coasts of western Europe (*TXx*), Scandinavia and eastern Europe (both *TXx* and *TNx*). Observational uncertainty has been assessed with the HadEX3 dataset and summer extreme temperature trend patterns broadly agree

between the two datasets but for *TNx* over the eastern part of WA region. The strong *TNx* cooling observed with the BERK dataset is reduced and shifted eastward when using the HadEX3 dataset. Finally, we have found that the AMV has likely contributed to both dynamic and thermodynamic changes in extreme temperature, in particular over a broad central Europe region north of the Black Sea.

Dynamical adjustment provides a quick and cheap (computationally) observationally-based storyline approach to assess and quantify the role of atmospheric circulation as a driver of extreme events. Dynamical adjustment can be performed in both factual and counterfactual (the world without human influence) worlds, assuming that the counterfactual can be simply defined by removing a non-parametric trend to the observed climate surface variable under scrutiny (here *TX* and *TN*). Note that, in principle, the contribution of forced atmospheric trends can also be estimated by dynamical adjustment (Deser et al., 2016). Assuming that a forced atmospheric trend can be detected and robustly estimated, dynamical adjustment can be performed twice, by removing or not the *SLP* trend from the raw *SLP* data. Difference between the two results for the reconstructed surface variable gives an estimate of the contribution of the forced dynamic component. In the standard conditional approach used here, the hypothesis is that forced atmospheric circulation changes are undetectable and no detrending is performed on the *SLP* field. In this case, the above dual approach (performing the dynamical adjustment in both factual and counterfactual worlds for the surface variable, here *TX* or *TN*) allows to partition the extreme event temperature anomaly in four contributions: the (internal) dynamic component, the (internal) thermodynamic residual, the forced long-term trend thermodynamical changes and the contribution due to forced changes in other factors such as the mean horizontal advection of forced changes in temperature gradients or vertical advection.

The above dynamical adjustment decomposition can then be used to present the approach results from two different perspectives: forced versus internal or dynamic versus thermodynamic. Comparison with other model-based methods, for instance those using spectral nudging, can then be performed from one or the other perspective. For example, van Garderen et al. (2021) use the spectral nudging method (with the ECHAM6 atmospheric model and the NCEP reanalysis) to make attribution statements regarding the 2010 Russian heatwave (their table 2). They focus on the role of climate change on the heatwave amplitude in early August (domain-averaged anomaly ~10 °C according to their estimate, relative to a 1985–2015 climatology). Based on their model results, they estimate that the heatwave amplitude can be split in two contributions, ~8 °C due to internal variability and ~2 °C being anthropogenically-forced. Assuming that the early August period can be taken as the first two weeks of August, the dynamical adjustment approach using *SLP* from the NCEP reanalysis gives a region-averaged heatwave anomaly of 9.9 °C with 8 °C and 1.9 °C from the contributions of internal variability and anthropogenic forcing, respectively (here we use the same climatological period as van Garderen et al. (2021)). Interestingly, despite the fact that the two approaches rely on very different methodology and data, their results on the relative influence of internal variability and anthropogenic forcing on the region-averaged heatwave anomaly are remarkably similar.

The dynamical adjustment methodological framework proposed in this study provides a simple and practical approach to investigate and quantify the role of atmospheric circulation in specific extreme events as well as long-term changes in extreme indicators. Combined with model-based approaches, dynamical adjustment results can improve the understanding and interpretation of observed extreme events with minimal effort in term of computing. Application to other climate parameters (for example precipitation extremes) and regions will be pursued in future work.

## Appendix A

**Method parameter sensitivity tests**

Sensitivity tests have been performed to support the final choice of the dynamical adjustment code parameters. We focus here on two of the parameters: the number of subsampled analogues, $N_s$, and the number of iterations, $N_r$. For $N_s$, the focus is on the accuracy of the *SLP* fit for the early 2010 European cold spell. Several multi-day periods, including the January 1–8 period marked by the occurrence of very cold *TX* anomalies over Europe (Fig. 2a) are used to assess the quality of the *SLP*

reconstruction. The dynamical adjustment code is run for a range of $N_s$ values ($N_s = 20, 40, 60 \ldots260, 280, 300$) keeping the other parameter values to those indicated in the main text. The metric is simply the root-mean square error (RMSE) between the daily original *SLP* and the reconstructed *SLP*. The RMSE is estimated separately for each day and the total RMSE is calculated as the sum in quadrature of the daily RMSE values. Figure A1 shows that the error is systematically large for small $N_s$ values (often greater than 2 hPa for less than 50 analogues), strongly decreases after $N_s \sim 100$ (to less than 1 hPa)

and almost saturates to a few tenths of hPa after $N_s \sim 200$ (with RMSEs less than 0.5 hPa beyond $N_s \sim 200$). The shape of the RMSE curve is very similar among the different periods. Note that $N_s$ is a key parameter in term of computational cost as it defines the matrix size involved in the calculation of the Moore-Penrose pseudo-inverse (see Appendix of Deser et al. 2016 for details). Therefore, the choice of $N_s \sim 200$ is a good compromise between accuracy and speed.

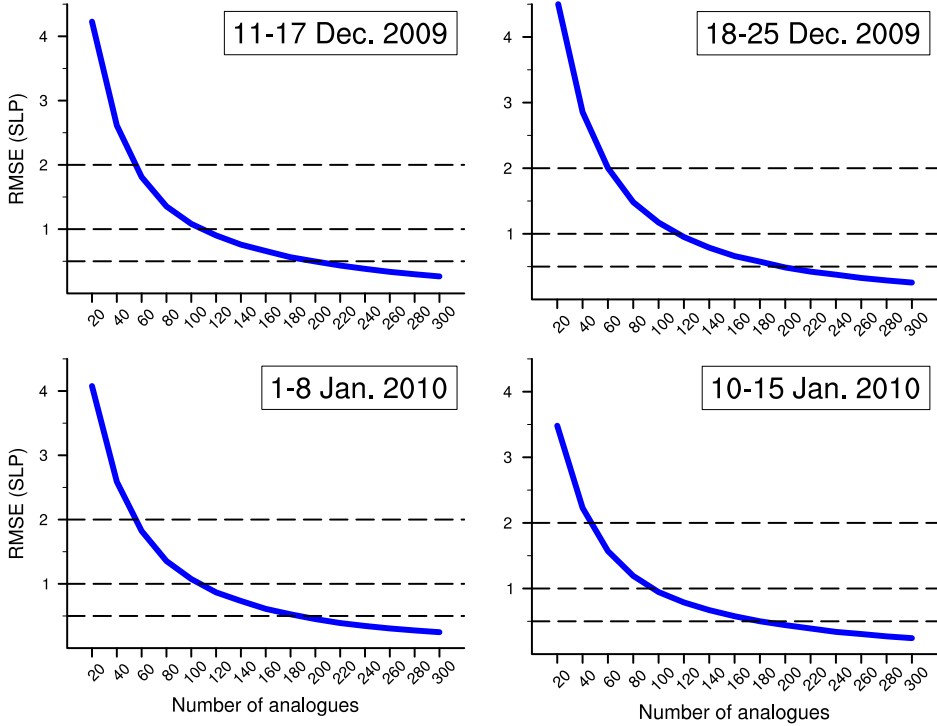


**Figure A1:** Accuracy (assessed by the RMSE, unit: hPa) of the *SLP* reconstruction as a function of the number of analogues used in the dynamical adjustment multilinear regression step. Four multi-day periods (dates in upper-right corner) within the 2010 early winter European cold spell are considered to perform the sensitivity test. The RMSE is first estimated at each grid-point and then averaged over the 30°N–75°N; 25°W–40°E geographical box with latitudinal weighting.

The second parameter is the number of iterations, $N_r$. In this case, the focus is on the reconstructed *TX*. The sensitivity test is performed for several multi-day periods throughout the Russian heat wave. The dynamical adjustment code is run for a range of $N_r$ values ($N_r$ = 10, 20, 30, … 150, 160 and with other parameters at their optimized value) and the metric measures the RMSE between the reconstructed *TX* for the different values in the above $N_r$ range and the reconstructed *TX* field with $N_r$ = 300, taken here as the "reference" value. Note that this only allows to see the "convergence" of the algorithm relative to the

number of iterations for a given value of the $N_a$ and $N_s$ parameters (here $N_a \sim$ 400 and $N_s \sim$ 200). Figure A2 shows that there is an initial fast RMSE decrease (starting from RMSE values of $\sim$ 1 °C for less than 10 iterations) followed by a slower convergence of the algorithm with the number of iterations. The change in convergence rate occurs when $N_r$ exceeds 50–60 iterations making the choice of $N_r \sim$ 100 a reasonable trade-off.

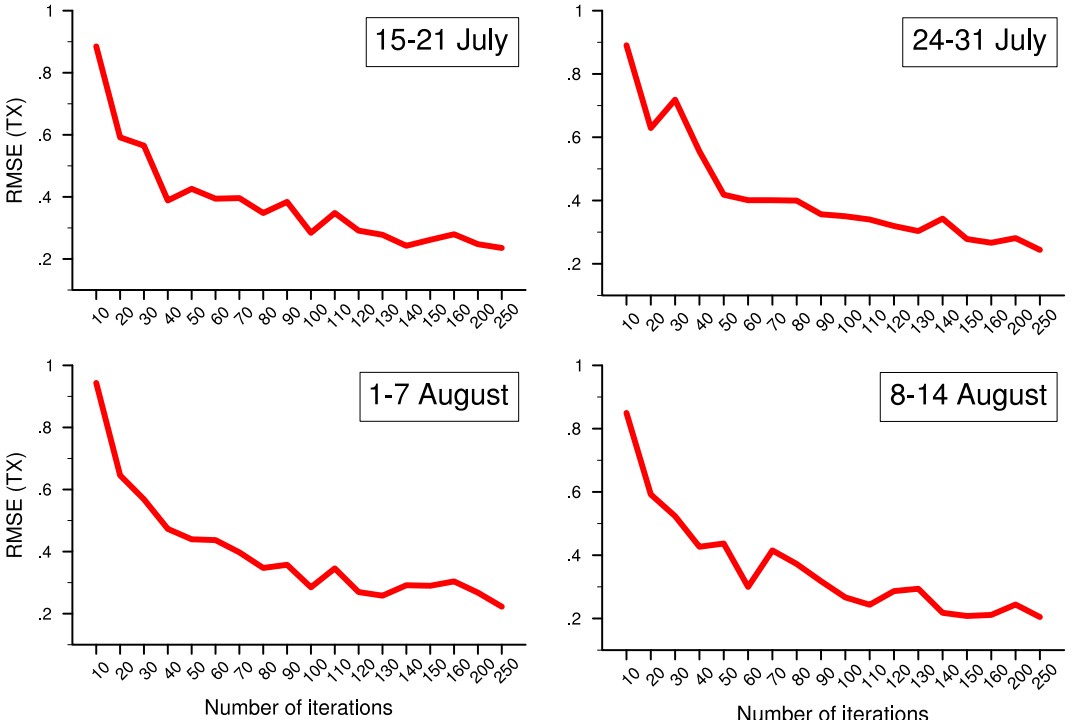

**Figure A2:** Convergence (assessed by the RMSE, unit: °C) of the *TX* reconstruction as a function of the number of iterations used in the dynamical adjustment algorithm. Convergence is assessed against a reference dynamical adjustment run performed with 300 iterations. Four multi-day periods (dates in upper-right corner) within the 2010 summer Russian heatwave are considered to perform the sensitivity test. The RMSE is first estimated at each grid-point and then averaged over the 35°N–65°N; 25°E–50°E geographical box with latitudinal weighting.

## Appendix B

**Time distribution of selected analogues**

The selected study period (1900–2018) used for the *SLP* analogue search includes active phases of low-frequency internal variability modes such as the Atlantic multidecadal variability (AMV). In addition, the 20CR_V3 reanalysis could possibly exhibit lower variance in the data-poor early period leading to a preferential selection of analogues from recent decades. Drawing a majority of analogues from a specific period can potentially bias the estimation of the dynamical component contribution and that of the internal residual. Figure B1 shows the distribution of analogues with respect to the years for the two extreme events (for the entire selected event, the total number of analogues used is equal to $N_r$ x $N_s$ x $N_d$, with $N_r$ and $N_s$ defined as in section 2.2 and $N_d$ the number of days of the event). It clearly shows that the selected sample of analogues does not favor any specific period nor exhibit any particular trend and that specific years with a large number of analogues can be found throughout the entire period.

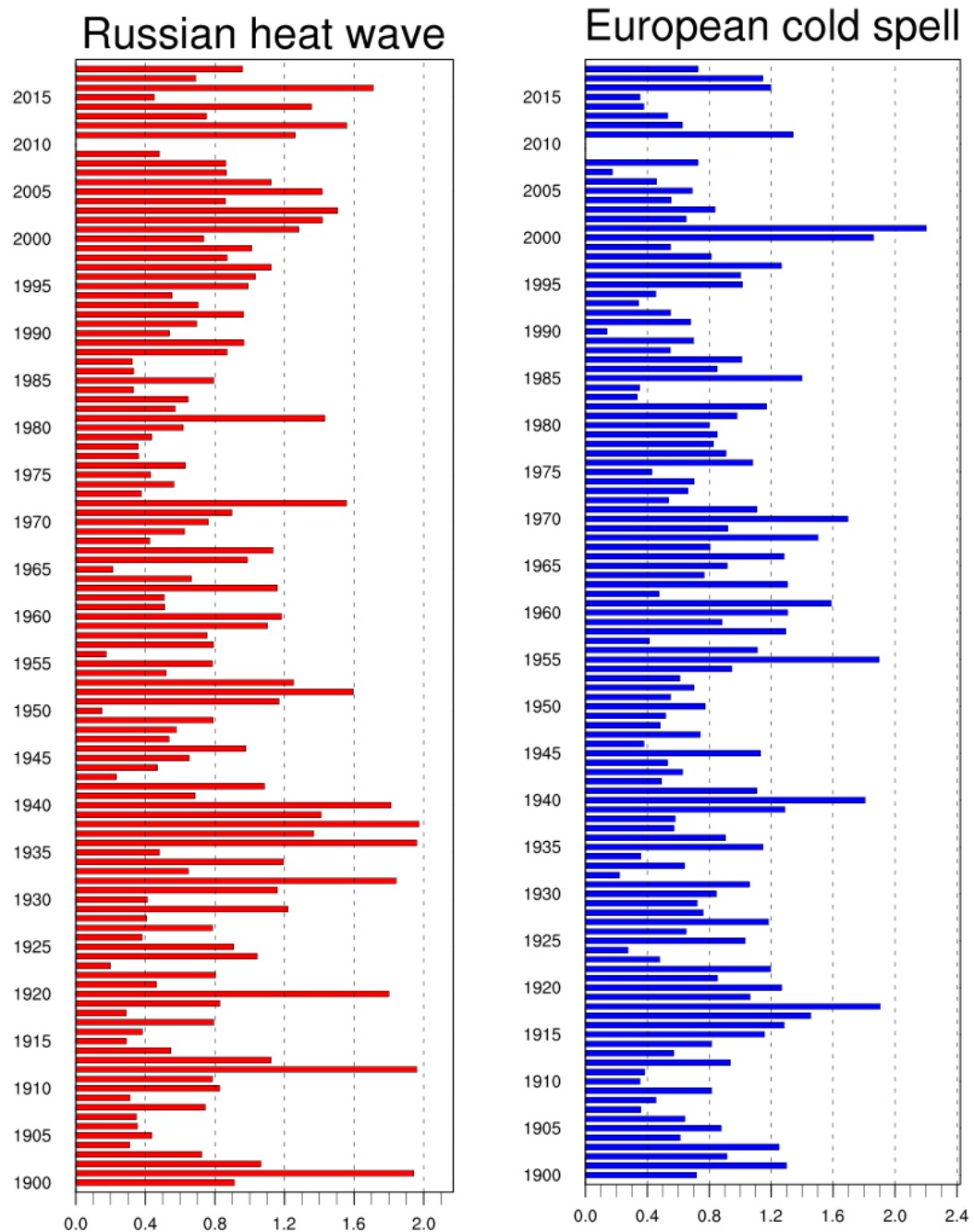

**Figure B1**: time distribution of selected *SLP* analogues (X-axis, unit in percent of the total number of used analogues) versus their year of occurrence (Y-axis) for the two extreme events

## Appendix C

**Observational uncertainty for the *TXx* and *TNx* trend analysis**

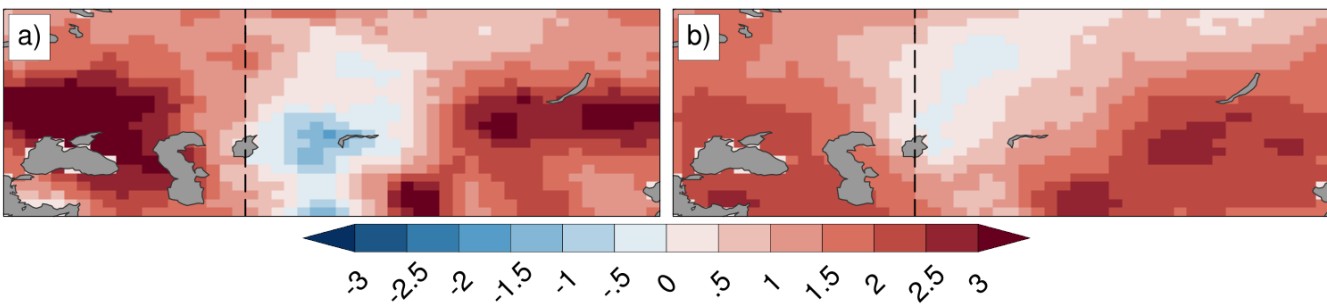

**Figure C1**: 1979–2018 temperature trend patterns (units: °C 40 yrs$^{-1}$) for western and central Asia based on the HadEX3 dataset: a) *TXx* and b) *TNx*. The black dashed line indicates the eastern boundary of the region map shown in Fig. 5.


**Code and data availability:**

The dynamical adjustment code used in this study is available on https://github.com/terrayl/Dynamico. Observed and reanalysis data used for this study can be found on the data provider websites. They can also be provided by the author upon request.


**Competing interest:**

The author declares no competing interests.

**Author contribution:**

LT developed the ideas, designed the methodology, developed the computer algorithms, analysed the data, created the figures, and wrote the manuscript.

**Acknowledgements**: Support for the Twentieth Century Reanalysis Project version 3 dataset is provided by the U.S. Department of Energy, Office of Science Biological and Environmental Research (BER), by the National Oceanic and
Atmospheric Administration Climate Program Office, and by the NOAA Physical Sciences Laboratory. The National Center for Atmospheric Research (NCAR) command language (NCL), version 6.5.0, has been used for all the dynamical adjustment computations and visualization of the results. L.T thanks Kathrin Wehrli and Tim Woollings for their insightful and constructive comments that improved the quality of the manuscript, and the co-editor, Thomas Birner, for his careful reading of the manuscript and efficient handling of the review process.

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
