# Peer review of "A dynamical adjustment perspective on extreme event attribution"

_Weather and Climate Dynamics, 2021_

## Author Comment (AC1)

**A dynamical adjustment perspective on extreme event attribution**
Response to reviewers

**Reply to reviewer 1** (K. Wehrli):

In the following the reviewer comments appear in black with the author responses in blue.
All the references mentioned in the text are given at the end.

**General Comments**

The manuscript is well-written, follows a clear narrative and the conclusions are supported by the analysis and literature cited in the paper. I especially appreciate the comprehensive literature review and that the observation-based results are frequently compared to results from modelling studies and vice versa. The study complements existing work in the field and provides added value to the understanding of past extreme events and long-term changes in extreme indicators. It shows that the dynamical adjustment method is a practical and versatile approach that can even be considered for rapid attribution of extreme events.

I thank the referee for her detailed lecture, helpful comments and suggestions, and appreciation of the manuscript.

I was wondering whether there is similar year-to-year variability in the contribution of the dynamic component to changes in TXx and TNx as there is on a daily basis for the specific events. Below I also added a comment about this point. Maybe the author has already done some analysis in this direction that he can share.

This is an interesting question. The interannual variability of the TXx and TNx changes induced by the dynamical component is substantial and comparable with the daily variability of dynamically-induced changes during extreme events. For instance, Figure R1 shown below suggests that the interannual variability of the dynamic component contribution to TXx changes over western Russia has a magnitude on the order of 2-3 °C, with larger values over the northern and central part of the domain. It is noteworthy that the interannual variability of TXx residual (thermodynamic) changes is slightly larger than that of the TXx dynamically-induced changes. The results are similar for TNx, albeit with smaller amplitudes (not shown). Figure R2 (right panel) shows the daily variability of the dynamically-induced TX changes during the 2010 Russian heatwave (from July 15 to August 14). The daily variability pattern shows very similar amplitudes to the interannual variability one shown in Figure R1c.

[Figure]

Figure R1: Summer interannual standard deviation over the 1979–2018 period: (a) Raw TXx (b) Thermodynamic contribution to TXx changes (c) Dynamic contribution to TXx changes

[Figure]

Figure R2: Daily standard deviation during the 2010 summer Russian heatwave: (a) Raw TX (b) Dynamic contribution to daily TX changes

**Minor comments**

l.37: The second approach of extreme event attribution is introduced as the "process-based or storyline approach". To me, the term «storyline» might need a little more explanation in this context as I think I would not name every process-based study a storyline. Both are not probabilistic and aim to understand the driving factors. However, following e.g. Shepherd et al. 2018, storylines also explore different plausible climates, which is not done in Wehrli et al. 2019 or the present study. I am aware that it is probably not possible to make a clear distinction between a process-based or storyline study in every case. It might help if the author could share his definition of a storyline.

Thank you for the comment. I do agree that the two expressions (process-based and storyline) are not strictly equivalent. Based on Shepherd et al. (2018), the storyline approach can be defined as a physically self-consistent unfolding of past events, or of plausible future events. For example, they write: "However, we also include past events, because historical events are not simply single data points but involve detailed stories which can be unpacked".

I would suggest that both the Terray and the Wehrli et al. papers are exactly doing that, the "unpacking" of past events. Perhaps, one can propose that the process-based approach is a sub-category of the broader storyline approach whose main focus is on past events. In a sense, one could say that the process-based approach also explores plausible **past** climates. For instance, from the Wehrli et al. results, it is possible to make some plausible inference on the Russian heatwave characteristics, had the soil moisture been at its climatological state (instead of being in a dry state).

I propose to leave only "process-based approach" on line 37 and to add the following sentence at the end of the paragraph: "Note that the process-based approach can also be viewed as a sub-category of the storyline approach that focuses on the key drivers and physically plausible unpacking of past events (Shepherd et al., 2018)."

l.60: I would leave away the word "tropical" as the heatwaves examined in Wehrli et al. 2019 were not really tropical events even if parts of the regions examined for Australia and South Africa can be classified as tropical/sub-tropical climate.

Yes, agreed. "tropical" has been replaced by "subtropical"

l.98: It would be nice to briefly mention what method was used in Horton et al. 2015 as the study will be referenced also later in the manuscript.

I have added this at the beginning of the sentence: "Based on a trend analysis of atmospheric circulation patterns derived from a self-organizing map clustering approach, Horton et al. ….

l.278: Do you mean "more persistent" instead of "intense"? The first spell in mid-December looks more intense to me than the second.

Yes, thank you. "intense" has been replaced by "persistent".

l.397: Could this trend in TX due to dynamics be related to one extreme event in the recent years such as the Russian heatwave? Would the maps look different if you left out 2010? And is there a lot of year-to-year variability in the contribution by dynamics?

The TXx trend patterns with year 2010 left out are shown in Figure R3. The patterns are very similar to the ones shown in the paper (Figure 5). This suggests that the TXx trend does not depend on the presence of 2010. As shown in Figure A1 and discussed above, the year-to-year variability in the dynamical contribution of TXx changes is quite substantial. A short discussion regarding these two points has been added to section 4.

[Figure]

Figure R3: Similar as Figure 5 of the paper but without year 2010

l.401: Do you have a hypothesis why the thermodynamic component trend is overall cooling TNx? Would you expect this to be due to the local or the remote contribution associated with internal variability?

Thank you for the question. First, I was wondering whether the sign (and amplitude) of the TNx raw trend observed in the eastern part of the domain was realistic. Therefore, I have used the HadEX3 dataset (gridded values of both TXx and TNx, Dunn et al. 2020) to check the BERK results. Figure R4 below shows the TXx and TNx 1979-2018 trends based on HadEX3 (**note that I have slightly extended the domain eastward in Figure R4 compared with Figure 5 in order to make the point below**). Figure R4 does show a reasonably similar pattern to that of BERK for the TXx trend pattern. However, the HadEX3 pattern is different from the BERK one for TNx, with weak warming in the center and eastern part of the domain shown in Figure 5 (or Figure R3). A slight cooling is observed east of the region shown in Figure 5 (or R3). In addition, the HadEX3 TXx and TNx trend patterns both show cooling east of 60°E in the western Siberia plains (with a larger cooling for TXx than TNx). This is in good agreement with recent findings about observed mean temperature changes in this region that have resulted from increased winter and spring precipitation in recent decades, delayed snowmelt and increased soil moisture in summer as well as land-atmosphere interaction (see Sato and Nakamura 2019, Bulygina et al. 2009, Bulygina et al. 2011, Guo et al. 2019).
Therefore, I speculate that the westward-shifted cooling seen in the BERK TNx trend pattern might result from the scarcity of stations in this region for the BERK TN dataset and the use of a

large influence ratio in the infilling procedure. I have added a discussion in section 4 about the TNx trend results and observational uncertainty, alluding to the HadEX3 figure that has been added to the appendix.

**TXx 1979-2018    TNx 1979-2018**

[Figure]

Figure R4: TXx and TNx 1979–2018 trend patterns derived from the HadEX3 dataset

**Details, typing errors, etc.**
To ensure reproducibility of the study the location and extent of the SLP regions that were used for the two events in 3.1 and 3.2 still need to be specified. I only found the numbers for the TX regions in the tables.

Geographical boundaries for SLP and TX have been added in the method section. For instance, SLP domains are the following:
Western Europe cold spell: 25°N–90°N; 60°W–100°E
Russian heat wave: 25°N–90°N; 20°W–80°E
Note that the numbers given in table legends correspond to the boundaries of the black boxes in Figure 1c and 3c.

I.103: Do you mean TN maxima?
Yes, thank you. "TN minima" has been replaced by "TN maxima".

L.115: Later in the manuscript (e.g Table 1) an underscore is used before the version number in 20CR_V3
The acronym 20CR_V3 is now used throughout the manuscript.

I.220: The period chosen for the cold European winter is not exactly two weeks but 17 days
"two-week period" has been replaced by "seventeen-day period".

I.237: Do you mean Fig. 1b?
Yes, thank you. The typo has been corrected.

l.254: insert "the" before amplitude
'the" has been inserted before amplitude.

Fig.1: I find it hard to distinguish the one contour line that is thicker. Line thickness could be increased or the zero SLP anomaly line could be highlighted in a different color e.g. violet.
The thickness of the 0-line has been increased.

l.269: typo, should be 2009-20**1**0 early winter
Thank you, the typo has been corrected.

l.270: Shouldn't the numbers in the brackets be -3.07 °C and -2.04 °C?
Yes, thank you, the numbers have been corrected.

l.271: 20CR_V3
20CR_V3C has been changed to "20CR_V3".

l.336: I would close the first bracket after "anomaly"
A bracket has been added after "anomaly".

l.381: For the TX, TN and the SLP domain it should say °E instead of °W.
Yes, thank you. The typos have been corrected.

l.387: I think the bracket saying "smallest" should be omitted as you are using the largest anomalies whether it is TX or TN.
Yes, thank you. "(smallest)" has been removed.

l.444: Was it not mitigated by around one third from -3.07°C to -2.04°C (instead of 50%)?
Yes, thank you for pointing this. It has been corrected.

**References**

Bulygina, O.N. et al., 2011 *Environ. Res. Lett.* 6 045204

Bulygina, O.N. et al., 2009 *Environ. Res. Lett.* **4** 045026

Dunn, R. J. H., Alexander, L. V., Donat, M. G., Zhang, X., Bador, M., Herold, N., et al. (2020). Development of an updated global land in situ-based data set of temperature and precipitation extremes: HadEX3. *Journal of Geophysical Research: Atmospheres*, 125, e2019JD032263. https://doi.org/10.1029/2019JD032263

Guo, R., Deser, C., Terray, L., & Lehner, F. (2019). Human influence on winter precipitation trends (1921–2015) over North America and Eurasia revealed by dynamical adjustment. Geophysical Research Letters, 46. https://doi.org/10.1029/2018GL081316

Sato, T., Nakamura, T. Intensification of hot Eurasian summers by climate change and land–atmosphere interactions. *Sci Rep* 9, 10866 (2019). https://doi.org/10.1038/s41598-019-47291-5

---

## Author Response (AR1)

**Revision of wcd-2021-40: "A dynamical adjustment perspective on extreme event attribution" by Laurent Terray Response to reviewers**

I would like to thank the two reviewers for their detailed reading, constructive comments and suggestions as well as their appreciation of the manuscript. In response to the main issues raised by the reviewers, I have made the following main changes:

- As suggested by reviewer 2, an uncertainty range of the dynamical component, based on bootstrapping, has been added to Figs. 1 and 3. This has led to a more robust assessment of the internal residual significance. The homogeneous distribution of analogues over time has also been verified (new Appendix B, with one figure).
- As asked by both reviewers, a detailed discussion of some of the causal factors behind the results shown in Figs. 5 and 6 has been added to section 4. The influence of the AMV is now being discussed and illustrated with Fig. 7 (see also below).
- The robustness of raw TX and TN trends (with a specific focus on the TNx trend displayed in Fig. 5) has been assessed with another dataset (HadEX3, Dunn et al. 2020). An additional figure (Fig. 7) has been added to support the analysis and discussion as well as a new Appendix, Appendix C, with one figure.

I have also taken into account all minor comments and typos. Note that I have also homogenized as much as possible the size and aspect of the label bar in figures 1, 3, 5, 6, 7.

In the following, the reviewer comments appear in black with the author responses in blue. All the references mentioned in the text are given at the end. The indicated lines refer to the tracked change manuscript.

**Reply to reviewer 1 (K. Wehrli):**

**General Comments**

The manuscript is well-written, follows a clear narrative and the conclusions are supported by the analysis and literature cited in the paper. I especially appreciate the comprehensive literature review and that the observation-based results are frequently compared to results from modelling studies and vice versa. The study complements existing work in the field and provides added value to the understanding of past extreme events and long-term changes in extreme indicators. It shows that the dynamical adjustment method is a practical and versatile approach that can even be considered for rapid attribution of extreme events.

I thank the referee for her detailed lecture, helpful comments and suggestions, and appreciation of the manuscript.

I was wondering whether there is similar year-to-year variability in the contribution of the dynamic component to changes in TXx and TNx as there is on a daily basis for the specific

events. Below I also added a comment about this point. Maybe the author has already done some analysis in this direction that he can share.

This is an interesting question. The interannual variability of the TXx and TNx changes induced by the dynamical component is substantial and comparable with the daily variability of dynamically-induced changes during extreme events. For instance, Figure R1 shown below suggests that the interannual variability of the dynamic component contribution to TXx changes over western Russia has a magnitude on the order of 2-3 °C, with larger values over the northern and central part of the domain. It is noteworthy that the interannual variability of TXx residual (thermodynamic) changes is slightly larger than that of the TXx dynamically-induced changes. The results are similar for TNx, albeit with smaller amplitudes (not shown). Figure R2 (right panel) shows the daily variability of the dynamically-induced TX changes during the 2010 Russian heatwave (from July 15 to August 14). The daily variability pattern shows very similar amplitudes to the interannual variability one shown in Figure R1c.

Figure R1: Summer interannual standard deviation over the 1979–2018 period: (a) Raw TXx (b) Thermodynamic contribution to TXx changes (c) Dynamic contribution to TXx changes

Figure R2: Daily standard deviation during the 2010 summer Russian heatwave: (a) Raw TX (b) Dynamic contribution to daily TX changes

**Minor comments**

I.37: The second approach of extreme event attribution is introduced as the "process-based or storyline approach". To me, the term «storyline» might need a little more explanation in this context as I think I would not name every process-based study a storyline. Both are not probabilistic and aim to understand the driving factors. However, following e.g. Shepherd et al. 2018, storylines also explore different plausible climates, which is not done in Wehrli et al. 2019 or the present study. I am aware that it is probably not possible to make a clear distinction between a process-based or storyline study in every case. It might help if the author could share his definition of a storyline.

Thank you for the comment. I do agree that the two expressions (process-based and storyline) are not strictly equivalent. Based on Shepherd et al. (2018), the storyline approach can be defined as a physically self-consistent unfolding of past events, or of plausible future events. For example, they write: "However, we also include past events, because historical events are not simply single data points but involve detailed stories which can be unpacked". I would suggest that both the Terray and the Wehrli et al. papers are exactly doing that, the "unpacking" of past events. Perhaps, one can propose that the process-based approach is a sub-category of the broader storyline approach whose main focus is on past events. In a sense, one could say that the process-based approach also explores plausible **past** climates. For instance, from the Wehrli et al. results, it is possible to make some plausible inference on the Russian heatwave characteristics, had the soil moisture been at its climatological state (instead of being in a dry state).

**Manuscript change:**

I propose to leave only "process-based approach" on line 37 and to add the following sentence at the end of the paragraph: "Note that the process-based approach can also be viewed as a sub-category of the storyline approach that focuses on the key drivers and physically plausible unpacking of past events (Shepherd et al., 2018)."

1.60: I would leave away the word "tropical" as the heatwaves examined in Wehrli et al. 2019 were not really tropical events even if parts of the regions examined for Australia and South Africa can be classified as tropical/sub-tropical climate.

Yes, agreed. Manuscript change: "tropical" has been replaced by "subtropical"

I.98: It would be nice to briefly mention what method was used in Horton et al. 2015 as the study will be referenced also later in the manuscript.

Yes, agreed.

Manuscript change:

I have added this at the beginning of the sentence: "Based on a trend analysis of atmospheric circulation patterns derived from a self-organizing map clustering approach, Horton et al. ....

I.278: Do you mean "more persistent" instead of "intense"? The first spell in mid-December looks more intense to me than the second.

Yes, thank you. Manuscript change: "intense" has been replaced by "persistent".

I.397: Could this trend in TX due to dynamics be related to one extreme event in the recent years such as the Russian heatwave? Would the maps look different if you left out 2010? And is there a lot of year-to-year variability in the contribution by dynamics?

The TXx and TNx trend patterns with year 2010 left out are shown in Figure R3. The patterns are very similar to the ones shown in the paper (Figure 5). This suggests that the trends do not significantly depend on the presence of 2010. As shown in Figure R1 and discussed above, the year-to-year variability in the dynamical contribution of TXx changes is quite substantial.

Manuscript change:

A short discussion regarding these two points has been added to section 4.